

# Towards a process-oriented understanding of the impact of stochastic perturbations on the model climate

Moritz Deinhard[1] and Christian M Grams[1]

[1]Institute of Meteorology and Climate Research (IMK-TRO), Department Troposphere Research, Karlsruhe Institute of Technology (KIT), Karlsruhe, Germany

**Correspondence:** Moritz Deinhard (moritz.deinhard@gmail.com)

**Abstract.**

Stochastic parametrisation techniques have been used by operational weather centres for decades to produce ensemble forecasts and to represent uncertainties of the forecast model. Their use has been demonstrated to be highly beneficial, as it increases the reliability of the forecasting system and reduces systematic biases. Despite the random nature of the perturba-
tion techniques, the response of the model can be nonlinear and the mean state of the model can change. In this study, we attempt to provide a process-based understanding how stochastic model perturbations affect the model climate. Previous work has revealed sensitivities of the occurrence of diabatically driven, rapidly ascending air streams to the stochastically perturbed parametrisation tendencies (SPPT) scheme. Such strongly ascending air streams are linked to different weather phenomena, such as precipitation and upper-tropospheric ridge building in the midlatitudes, which raises the question whether these pro-
cesses are also influenced by stochastic perturbations.

First, we analyse if rapidly ascending air streams also show sensitivities to a different perturbation technique - the stochastically perturbed parametrisations (SPP) scheme, which directly represents parameter uncertainty in parametrisations and has recently been developed at the European Centre for Medium-Range Weather Forecasts (ECMWF). By running a set of sen-
sitivity experiments with the Integrated Forecasting System (IFS) and by employing a Lagrangian detection of ascending air streams, we show that SPP results in a systematic increase of the occurrence of ascending air parcel trajectories compared to unperturbed simulations. This behaviour is very similar to that of SPPT, albeit some regional differences are apparent. We further show that the one-sided response to the stochastic forcing cannot be attributed to a single process (e.g. convection parametrisation), but rather that perturbations to different parametrisations have similar effects.

Thereafter, we link the frequency changes of ascending air streams to closely related weather phenomena. Whereas the sig-nal of increased ascending motion is directly transmitted to global precipitation sums for all analysed schemes, changes to the amplitude of the upper-level Rossby wave pattern are more subtle. In agreement with the trajectory analysis, both SPPT and SPP increase the waviness of the upper-level flow and thereby reduce a systematic bias of the model, even though the order of
magnitude is small.





Our study presents a coherent process chain that enables to understand how stochastic perturbations systematically affect the model climate. We argue that weather systems which are characterised by threshold behaviour on the one hand and that serve as a dynamical hinge between spatial scales on the other hand can convert zero-mean perturbations into an asymmetric

response and project it onto larger scales.

## 1 Introduction

Ensemble prediction aims to represent forecast uncertainty by estimating the future probability density function of the atmospheric state (Leutbecher and Palmer, 2008). This is typically achieved by running multiple integrations of the forecast model with small perturbations which grow with forecast lead time due to the chaotic behaviour of the atmosphere and thereby repre-

sent different realisations of the evolution of the system. During the early years of ensemble prediction, probabilistic forecasts were generated with perturbations to the initial conditions only (Lewis, 2005). Such forecasts, however, are underdispersive and on average do not adequately capture the uncertainty of the forecast (e.g. Palmer et al. 2005). To further increase the dispersion of the ensemble members, techniques that represent uncertainty related to the forecast model have been developed and implemented into operational systems, and are still in use today (Leutbecher et al., 2017).


Model errors are largely related to the spatial discretisation of numerical weather prediction (NWP) models, which requires the representation of processes on the sub-grid scale through parametrisations. A common approach to address model error is therefore to represent uncertainties in the parametrisation schemes. For example, the stochastically perturbed parametrisation tendencies (SPPT) scheme randomly perturbs the net tendencies from all parametrised processes (Leutbecher and Palmer,

2008), and has been operational at the European Centre for Medium-Range Weather Forecasts (ECMWF) since 1998 (Buizza et al., 1999). The incorporation of SPPT into the operational forecast chain resulted in increased ensemble spread (mainly in the tropics, but also in the Extratropics) and thereby led to a more reliable ensemble, and improved the probabilistic skill of the forecasting system (Leutbecher et al., 2017). Despite these undoubtedly advantageous properties, it has been shown that some of the basic assumptions of SPPT are not well justified, such as the coherency of the perturbations for different parametri-

sations or for different prognostic variables (Leutbecher et al., 2017; Christensen, 2020). Therefore, ECMWF developed the stochastically perturbed parametrisations (SPP) scheme, which perturbs individual parameters in the parametrisations instead of the net tendencies from all schemes and thereby overcomes some of the limitations of SPPT (Ollinaho et al., 2017; Lang et al., 2021).

Several studies have shown that the advantages of stochastic parametrisation exceed a simple increase of ensemble spread (e.g. Berner et al. 2012, 2017). The use of stochastic parametrisation can, for example, reduce systematic model errors and biases, as shown for tropical precipitation (Weisheimer et al., 2014; Subramanian et al., 2017; Strømmen et al., 2019), ENSO (Christensen et al., 2017; Yang et al., 2019), or tropical cyclones (Stockdale et al., 2018; Vidale et al., 2021). Even though such effects have mainly been reported for tropical regions, where stochastic parametrisations are most active due to large



parametrisation tendencies (Leutbecher et al., 2017), positive impacts have also been found for the Extratropics, albeit of a more subtle nature. For example, stochasticity has been shown to reduce biases in the extratropical flow (Berner et al., 2012), to improve the representation of midlatitude circulation regimes in different model hierarchies (Dawson and Palmer, 2015; Christensen et al., 2015), to improve the representation of extratropical teleconnection patterns (Strømmen et al., 2022), and to influence atmospheric blocking (Berner et al., 2008; Davini et al., 2021).


In the literature, different pathways how random perturbations affect the mean state of the model have been discussed. Changes of the characteristics of large-scale extratropical weather regimes due to perturbations are commonly attributed to "noise-induced" drifts resulting from multiplicative forcing in non-linear systems (e.g. Sardeshmukh et al. 2001; Derbyshire et al. 2004; Birner and Williams 2008; Berner et al. 2012). From a smaller-scale, process-oriented perspective, Tompkins and

Berner (2008) show that positive humidity perturbations in the boundary layer are more effective in triggering convection than negative perturbations can suppress it. Likewise, Pickl et al. (2022) show that SPPT systematically increases the frequency of diabatically driven, rapidly ascending air streams and they suggest that zero-mean perturbations may result in a biased response when the system is characterised by a threshold behaviour.

Such rapidly ascending air streams, usually related to tropical convection and to slantwise ascent in the warm sector of extratropical cyclones (so-called warm conveyor belts (WCBs), Carlson 1980), are closely linked to different phenomena in the atmosphere. For example, both tropical convection as well as WCBs are associated with large amounts of precipitation on synoptic time scales (Jiang and Zipser, 2010; Pfahl et al., 2014). In the Extratropics, WCBs play an important role in shaping the large-scale circulation (e.g. Grams et al., 2011): the diabatically driven air stream transports lower-tropospheric air with low

values of potential vorticity (PV) into the upper troposphere, where higher values of PV prevail climatologically (e.g. Madonna et al. 2014). This diabatically generated negative upper-level PV-anomaly accelerates the upper-level jet by sharpening the PV-gradient along the tropopause (Grams et al., 2013), deflects the tropopause pole- and upward, contributes to ridge building and thereby amplifies the Rossby wave pattern (Pomroy and Thorpe, 2000; Grams and Archambault, 2016; Chagnon et al., 2013; Methven, 2015; Saffin et al., 2021). Eventually, this may lead to the formation and maintenance of blocking anticyclones

(Pfahl et al., 2015; Steinfeld and Pfahl, 2019). As WCBs and their impact on the large-scale flow are sensitive to the conditions provided by the background flow (e.g. low-level moisture supply (Schäfler and Harnisch, 2015) or baroclinicity (Grams et al., 2018)), WCBs act as a dynamical hinge between the lower and upper troposphere and between the synoptic and large scale, which makes WCBs very relevant for the growth and propagation of forecast errors (Martínez-Alvarado et al., 2016; Berman and Torn, 2019; Maddison et al., 2019; Pickl et al., 2023).


Building on the findings from Pickl et al. (2022), who found a systematic increase of the occurrence frequencies of rapidly ascending air streams with SPPT in ECMWF's ensemble prediction system, this study examines if the newly developed SPP-scheme results in a similar behaviour as SPPT. Also variants of SPP with perturbations to individual parametrisations will be considered. Subsequently, it will be investigated if the observed sensitivities to the SPPT- and SPP-scheme are reflected



in changes to related processes, i.e. precipitation and the waviness of the large-scale extratropical flow. To this end, we aim
at providing a process-level understanding how stochastic physics perturbations systematically affect the mean state of the
forecast model, and propose a coherent process chain.

The study is structured as follows: In section 2.1, we give an overview over the experimental setup and the model uncertainty
schemes that are investigated throughout the study. The approach how rapidly ascending air streams are detected is outlined
in section 2.2, followed by a detailed description how the amplitude of the upper-level Rossby wave pattern is assessed (sec-
tion 2.3). Additional data sets complementing the numerical experiments are described in section 2.4. In the results chapter,
sensitivities of the occurrence frequencies of rapidly ascending air streams to SPPT, SPP, and variants of SPP are discussed
in section 3.1. Thereafter, sensitivities of precipitation (section 3.2.1) and of the Rossby wave amplitude (section 3.2.2) to the
schemes are shown. Finally, the results are discussed and summarised in chapters 4 and 5.

## 2    Data and Methods

### 2.1    IFS ensemble experiments

We analyse the impact of different model uncertainty schemes on the rapidly ascending air streams and related processes using
a set of numerical experiments with the Integrated Forecasting System (IFS) CY46R1 of ECMWF. In this study, the SPPT-
and SPP-scheme are evaluated. The SPPT-scheme is the operational model uncertainty scheme at ECMWF and has been used
since 1998 (Buizza et al., 1999). It perturbs the model physics by multiplying the net tendencies from all parametrisations with
a random field which evolves in space and time (Leutbecher et al., 2017). Similarly to the SPPT-scheme, also SPP perturbs
the physical parametrisations. However, instead of the bulk approach of perturbing the net tendencies from all processes, a set
of selected parameters considered uncertain in the parametrisations of turbulent diffusion, orographic drag, convection, cloud-
and large-scale precipitation, and radiation are perturbed. For a detailed description of the SPP scheme, the reader is referred
to Lang et al. (2021). Additionally, we investigate two further simulations with parameter perturbations in selected parametri-
sation schemes: only in the convection scheme (SPP-CONV-ONLY) and in all parametrisation schemes, but the convection
scheme (SPP-CONV-OFF).

The initial conditions of all experiments are perturbed with an ensemble of data assimilations (Buizza et al., 2008; Isaksen
et al., 2010) and a singular vector technique (Leutbecher and Palmer, 2008). Hence, the experiments SPPT, SPP, SPP-CONV-
ONLY and SPP-CONV-OFF all feature both intial condition and model perturbations. To assess the impact of the perturbations,
a reference experiment with only initial condition perturbations, but without model perturbations is used (IC-ONLY). For each
of these experiments, 32 ensemble forecasts with 20 perturbed members and initialisations every other day between August
15th and October 15th have been run at a resolution of TCo399 (average grid spacing of 29 km) and 91 vertical levels until 12
days lead. The SPP sensitivity experiments (SPP-CONV-ONLY and SPP-CONV-OFF) have only been run for 11 initialisations
during the same period. For post-processing, the data is retrieved 6-hourly on a regular 1°x1° longitude-latitude grid.



## 2.2 Detection of rapidly ascending air streams

We use the software tool LAGRANTO (Wernli and Davies, 1997; Sprenger and Wernli, 2015) to compute offline trajectories
based on the 3-dimensional wind field outputted by the forecast model. The trajectories are started 6-hourly on a global 100 km
equidistant grid on 13 equally spaced pressure levels between 1000 and 700 hPa and computed forward in time until 48 h.
Subsequently, only those trajectories that ascend by at least 600 hPa within 2 days are retained and considered as 'rapidly as-
cending' (c.f. Pickl et al. (2022) for further information). We track the evolution of potential temperature along the trajectories,
which allows to compute the latent heating rate during the ascent of the air parcel.

## 2.3 Ridge and trough detection and Rossby wave amplitude

To assess the impact of the different model uncertainty schemes on the large-scale upper-level flow, we employ the technique
from Gray et al. (2014) to classify each grid point on an isentropic surface into one of four different categories 'trough', 'ridge',
'polar vortex' or 'subtropics'. This approach is based on the potential temperature-potential vorticity ($\theta$-PV) framework and
uses the dynamical tropopause, in this work defined as the 2 PVU contour on an isentropic surface ($PV_{tp}$), to determine the
structure of the upper-level flow. The dynamical tropopause is used to derive the equivalent latitude ($\Phi_{eq}$), which is defined as
the perimeter of a circle centered on the pole that encloses the same area as the instantaneous 2 PVU contour on an isentropic
surface (Butchart and Remsberg, 1986). This zonally symmetric background state encloses the same mass and circulation as
the full instantaneous PV-field (Methven and Berrisford, 2015), and can be interpreted as hemispheric-mean latitude of the
dynamical tropopause. To determine $\Phi_{eq}$, the sum of the areas of every grid point exceeding $PV_{tp}$ is computed individually for
each isentropic surface and valid time; the equivalent latitude is then obtained from the ratio of this area ($A_{PV_{tp}}$) to the area of
the whole hemisphere ($A_{hem}$) by

$$\Phi_{eq} = \arcsin(1 - \frac{A_{PV_{tp}}}{A_{hem}}). \tag{1}$$

For the classification procedure, the PV-value of each grid point is at first compared to $PV_{tp}$; subsequently, the latitude of
the grid point ($\Phi$) is compared to $\Phi_{eq}$. A grid point on the northern hemisphere is then classified as

 – 'trough', when $PV > PV_{tp}$ and $\Phi < \Phi_{eq}$

 – 'ridge', when $PV < PV_{tp}$ and $\Phi > \Phi_{eq}$.

When $PV > PV_{tp}$ and $\Phi > \Phi_{eq}$, the grid point is classified as 'polar vortex', and for $PV < PV_{tp}$ and $\Phi < \Phi_{eq}$, it is classi-
fied as 'subtropics'. The two latter categories are, however, not considered in this study and only listed here for completeness.
Note that we do not additionally consider cut-off lows as done in Gray et al. (2014), but classify such features into the category
'trough'. Figure 1 gives an illustration of the identification of upper-level troughs and ridges in a selected situation.





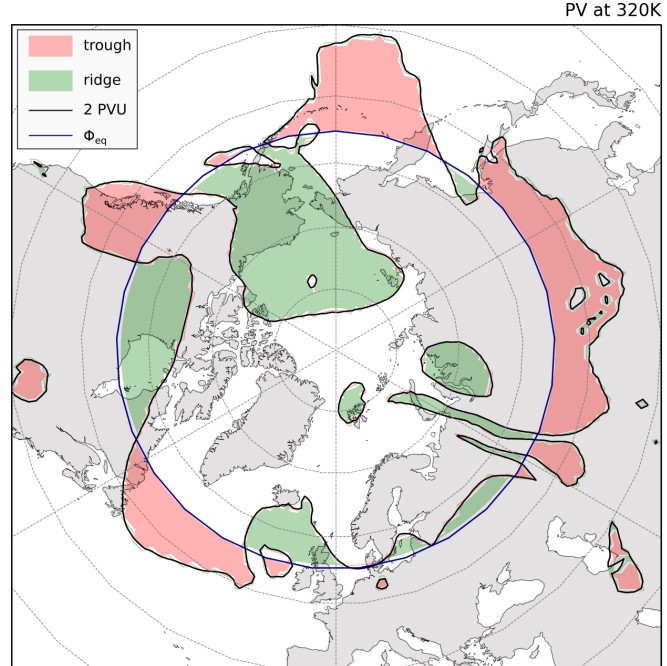

**Figure 1.** Identification of troughs (red) and ridges (green) on the 320 K isentrope based on the 2 PVU contour (black contour) in the experiment *SPPT* initialized on September 26 00 UTC 2016 at a forecast lead time of 7 days. The equivalent latitude $\Phi_{eq}$ is shown as blue line.

Subsequently, the hemispheric sum of the ridge and trough areas is computed at each valid time for each ensemble member on different isentropic levels. We use this as a proxy metric for the amplitude of the upper-level Rossby wave pattern: large ridge and trough areas correspond to a flow configuration where the upper-level waveguide is characterized by meridional
displacements. In turn, in a purely zonal flow configuration without any undulations of the waveguide, ridge and trough areas are minimised. Note that this technique does not provide any information about the depth (i.e. the strength of a PV-anomaly) of the Rossby wave pattern and does not consider the asymmetry that is usually observed for troughs and ridges.

## 2.4 Complementary data sets

### 2.4.1 Verification data

For verification purposes, two different datasets are employed. The WCB trajectory data (c.f. section 2.2) of the experiments is compared against ECMWF's operational high-resolution analysis (Rabier et al., 2000) interpolated to the grid of the ensemble experiments (ECMWF, 2019). The ERA5 reanalysis dataset (Hersbach et al., 2020) is employed as reference for the ridge and trough areas.





### 2.4.2 Operational ensemble forecasts

To increase the sample size for the Rossby wave amplitude analysis, we use ECMWF reforecast data from the subseasonal to seasonal (S2S) prediction project database (Vitart et al., 2017) initialized twice-weekly between 1997 and 2017 and compare the ridge and trough areas in the perturbed forecasts (SPPT + ICP) against the unperturbed control member. This is motivated by the insights from Pickl et al. (2022) who found that initial condition perturbations do not systematically affect the distribution of vertical velocities and the occurrence frequency of rapidly ascending air streams. The comparison of the perturbed and unperturbed members can therefore be used to evaluate the effect of the SPPT-scheme compared to unperturbed runs, which substantially increases the sample size. The reforecasts comprise 10 perturbed and one unperturbed member and are run at a spatial resolution of TCo639 (approx. 18 km). The dataset only contains PV at 320 K, such that the detection of ridges and troughs can only be done on one isentropic level. In total, we consider 3,200 initializations in winter (DJF, n=920), spring (MAM, n=1,060) and autumn (SON, n=1,040); summer is omitted, as the 320 K isentrope is not located in the upper-, but in the mid-troposphere and therefore does not adequately represented the Rossby wave structure.

The resolution of the archived reforecast data set is too coarse to compute trajectories as mentioned above. Therefore, we use an additional data set, consisting of operational ECMWF medium-range ensemble forecasts initialized twice daily (00 and 12 UTC) between Dec 2018 and Nov 2020 (2 years) archived regionally in the North Atlantic domain (15°–80° N, 130° W–80° E). For this data set, WCB trajectory data has been computed in the same way as described in section 2.2. As for the Rossby wave amplitude, the unperturbed control member is compared against the 50 perturbed (SPPT + ICP) ensemble members. This dataset will be consulted to determine seasonal differences of the effect of stochastic perturbations on the trajectories. Note that it is not possible to use this data set to compute the Rossby wave amplitude, as a hemispheric coverage of the data is required. Therefore, both the operational and the reforecast data sets have to be used in order to relate the trajectory analysis and the Rossby wave amplitude.

## 3 Results

### 3.1 Sensitivities of rapidly ascending air streams to different model uncertainty schemes

Following up on the results from Pickl et al. (2022), who for the first time show sensitivities of diabatically heated, rapidly ascending air streams to the SPPT-scheme, we here analyse whether also other model uncertainty schemes result in a similar behaviour as SPPT. Figure 2 shows the distribution of the number of trajectories in different experiments (colored bars) and in the verifying analysis (grey bars) in different regions. As reported in Pickl et al. (2022), the number of trajectories in the experiment with SPPT (red bars) is larger than in the experiment with initial conditions only (blue bars) in all regions, and this effect is more pronounced in the tropics than in the Extratropics. Interestingly, the experiment with SPP (yellow bars) shows a very similar behaviour as the one with SPPT: the trajectory counts are higher than in the unperturbed forecasts in all investigated regions. Comparing SPP to SPPT shows that the global trajectory count is slightly higher with SPP, which is predominantly driven by increased numbers in the tropics. In the North Atlantic and in the Northern Hemisphere Extratropics, in contrast, the



trajectory count is somewhat decreased in SPP compared to SPPT.

The experiments SPP-CONV-ONLY and SPP-CONV-OFF also show higher frequencies of rapidly ascending air streams
than the unperturbed experiment in all regions; remarkably, the effect is larger in the experiment with perturbations only to the
parameters in the convection parametrisations than in the experiment with perturbations in all parameters except for convection.
This indicates that the perturbations in the convection scheme are more efficient in triggering rapidly ascending air streams
than perturbations to all other parametrisations. In the Extratropics and the North Atlantic, the counts in SPP-CONV-ONLY
are even larger than in SPP, pointing towards the dominant role of perturbations in the convection parametrisation. The added
differences of the trajectory counts of the schemes SPP-CONV-ONLY and SPP-CONV-OFF to the unperturbed experiment
(IC-ONLY) are larger than the counts of SPP, indicating that the effects of the perturbations in different parametrisations are

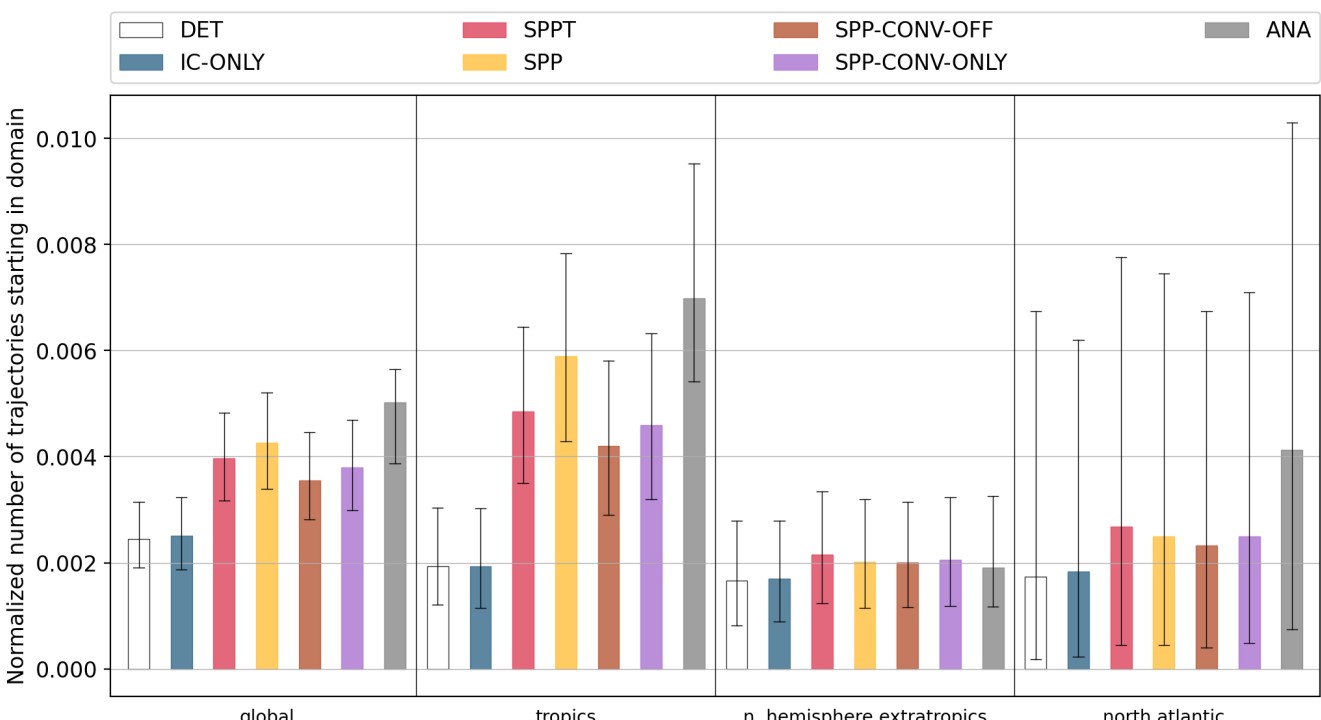

**Figure 2.** Counts of trajectories classified as rapidly ascending starting in the global domain, the tropics, the Northern Hemisphere Extra-
tropics, and the North Atlantic for the deterministic control run DET (transparent bar), the experiment with inital condition perturbations
only (IC-ONLY, blue), and the perturbed experiments SPPT (red), SPP (yellow), SPP-CONV-OFF (brown) and SPP-CONV-ONLY (purple).
Values of the verifying analysis (ANA) are shown by the grey bar. The counts are computed over all members (20), lead times (49) and initial
times (11 for experiments SPP-CONV-ONLY and SPP-CONV-OFF, 32 for all other experiments). Counts are normalized by the maximum
number of trajectories in each domain. The bar height displays the median and the whiskers the interquartile range.



partly superimposed or cancel each other.

The regional differences of the altered trajectory counts in the different experiments are largely controlled by the net latent
heat release along the ascending air stream: Figure 3 shows the ratio of the number of trajectories in the experiments with a
perturbed model (colored line) to the experiment with initial conditions only, as a function of the diabatic heating rate along
the trajectories (right axis). For additional orientation, the grey bars show the number of trajectories for each heating interval
started globally. The bimodal structure clearly shows two heating regimes (c.f. also Pickl et al. (2022)), with peak occurrence
frequencies in the heating range between 22.5-30 K (extratropical regime) and in the range between 40 and 50 K (tropical
regime). All experiments feature an exponential growth of the relative trajectory count with increasing latent heat release: for
heating rates between about 20 and 30 K, the ratios grow from about 1.05 to 1.25, with slightly larger values in SPPT compared
to SPP (which is also reflected in the trajectory counts in the North Atlantic and the Northern Hemisphere). At the transition
between the extratropical and tropical regimes, the ratios strongly increase and reach values larger than 2 for latent heating
rates between 40 and 50 K. In the tropical regime, the ratio is larger for SPP than for SPPT, which is also reflected in Figure

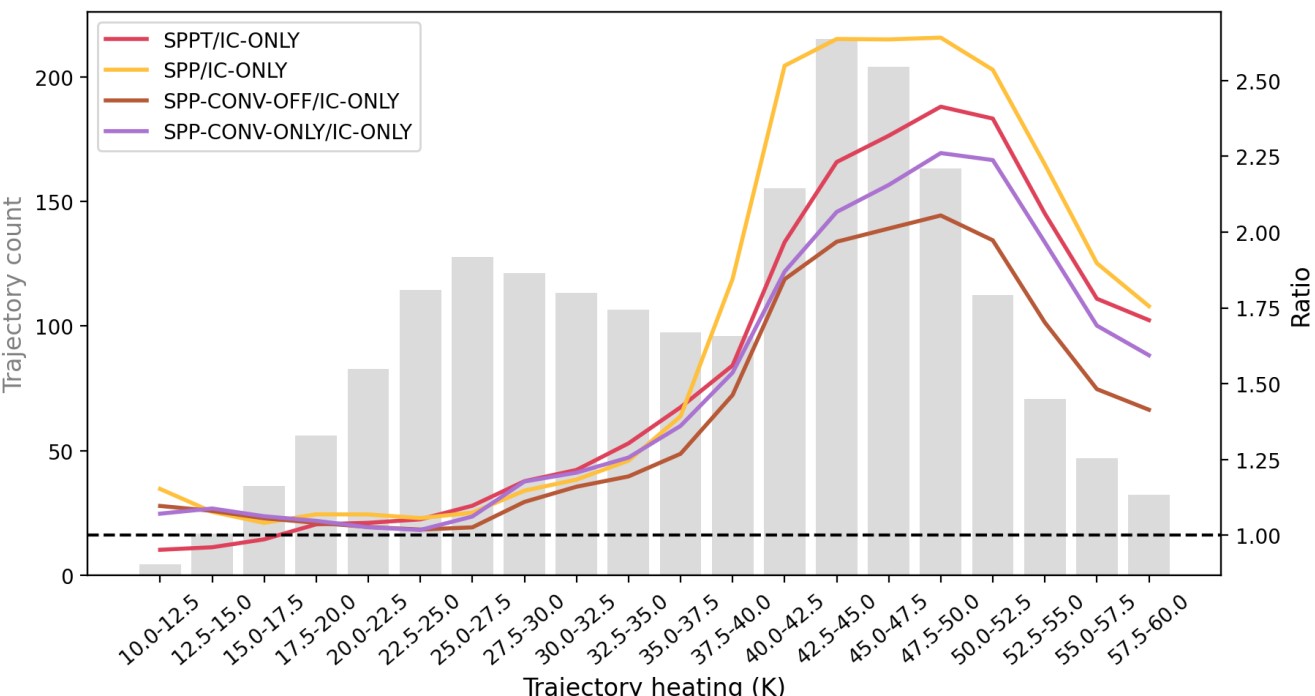

**Figure 3.** Number of trajectories started globally per 6-hourly time step in the experiment SPPT per integrated heating rate (gray bars, left
axis) and ratio of the trajectory counts in the experiments SPPT (red), SPP (yellow), SPP-CONV-OFF (brown), SPP-CONV-ONLY (purple)
and STOCDP (lightblue) and the unperturbed experiment IC-ONLY as function of the integrated latent heating rate along the trajectories.
The bin width is 2.5 K, and the ratios are only plotted for those heating rates that occur at least once per time step. Averaged over 11 forecasts.



2. Both SPP-CONV-ONLY and SPP-CONV-OFF follow a similar exponential behaviour, but the ratios are generally smaller than for SPP (except for the range between 22.5 and 32.5 K, where the curve of SPP-CONV-ONLY is slightly higher than that of SPP). For all heating rates larger than 25 K, perturbations to only the convection scheme result in a larger increase of rapidly ascending trajectories than perturbations to all other parametrisations, which is most pronounced in the tropical regime. The curve of SPPT lies below 1 for the lowest heating rates, indicating an inverse effect of SPPT on the rapidly ascending air streams which becomes relevant for weakly heated warm conveyor belts in wintertime.

The modified frequencies of rapidly ascending air streams through stochastic model perturbations, derived from a Lagrangian perspective, can also be analysed in an Eulerian framework. Figure 4 shows the differences of the number of grid points (i.e. difference histograms) associated with values of vertical velocities (i.e. $\omega$ at 500 hPa) between the experiments with perturbed model physics and the experiment with an unperturbed model. The negative value range on the x-axis corresponds to upward motion, whereas positive values represent downward motion. For all scales of vertical motion, model physics perturbations through SPPT and SPP result in qualitatively very similar changes of the occurrence of vertical velocities: in the upward spectrum of the histogram, SPPT and SPP are characterized by a structure with two maxima in the regions of very rapid ascents ($\omega < -0.4$ (SPPT) and $\omega < -0.6$ Pa/s (SPP), respectively) and of very slow ascents (-0.05 $< \omega <$ -0.2 (SPPT) and -0.05 $< \omega <$ -0.3 Pa/s (SPP), respectively), showing that vertical motions in these value ranges occur more often with SPPT/SPP than without model perturbations. This increase is compensated at the expense of moderate ascents (negative values in the range of -0.3 $< \omega <$ -0.6 Pa/s for SPPT and -0.2 $< \omega <$ -0.4 Pa/s) and very slow vertical velocities that are almost at rest (-0.05 $< \omega <$ 0 Pa/s). On the downward side of the spectrum, vertical velocities are uniformly accelerated through both SPPT and SPP, as shown by the decreased number of grid points with slow downward motions (0 $< \omega <$ 0.1 Pa/s) and an increased number of grid points with faster downward motions ($\omega > 0.1$ Pa/s). Qualitatively, the vertical motions are altered similarly by perturbations to only convection and by perturbations to all other parametrisations, but in line with the trajectory diagnostics, the effect on the largest velocities is weaker when the convection parametrisation remains unperturbed. Overall, the impact of the schemes on mid-tropospheric vertical velocities can be interpreted as an acceleration in two different regimes:

1. Fast updrafts are increased at the expense of moderate ascents (reflected in the increased number of rapidly ascending air streams obtained from the Lagrangian approach).

2. Air parcels at rest (i.e. with very small/negligible up- and downward velocities) are accelerated.

The increased upward mass flux is balanced by an uniform acceleration of downward velocities. Overall, SPPT and SPP amplify the vertical circulation of the atmosphere.

Even though the histograms of SPPT and SPP are very similar, there are some differences in the detailed structure of the occurrence frequencies: In the regime of the very rapid ascents, SPPT results in a more pronounced acceleration, as visible by the larger values and the broader range (x-axis intersect at about -0.4 Pa/s for SPPT and -0.6 Pa/s for SPP). This also results in a shift of the value range where the number of grid points is decreased compared to the unperturbed run. The good agreement



between the responses of both the rapidly ascending air streams detected by trajectory analysis as well as of the vertical mo-
tions from an Eulerian perspective between SPPT and SPP suggests that the underlying mechanism how the perturbations act
is similar for the two schemes.

The results presented so far show that the investigated stochastic model uncertainty representations all result in a similar uni-
directional response of the vertical velocities in the model, even though the introduced perturbations are random, symmetric
and zero-mean (Leutbecher et al., 2017; Lang et al., 2021). Pickl et al. (2022) argue that such a one-sided response could result
from meteorological processes which are characterised by a distinct threshold behaviour and are more likely to be triggered by
perturbations in one direction than to be suppressed by a perturbation of the same amplitude, but of opposite sign. Examples
for such processes are the triggering of atmospheric convection or the formation of clouds or precipitation. Therefore, they
concluded that "*other kinds of perturbation techniques [than SPPT] could also result in similar effects*" (Pickl et al., 2022).
Both SPPT and SPP introduce perturbations that are largest where parametrisations are active, for example in regions with
cloud formation or precipitation (Leutbecher et al., 2017). Diabatically driven rapid ascents are mainly located in such regions,
resulting in large perturbations that are likely to trigger more air parcels to rise than retain air parcels in the lower troposphere.
At this point, it would be interesting to also investigate perturbation schemes that represent other types of model error unrelated

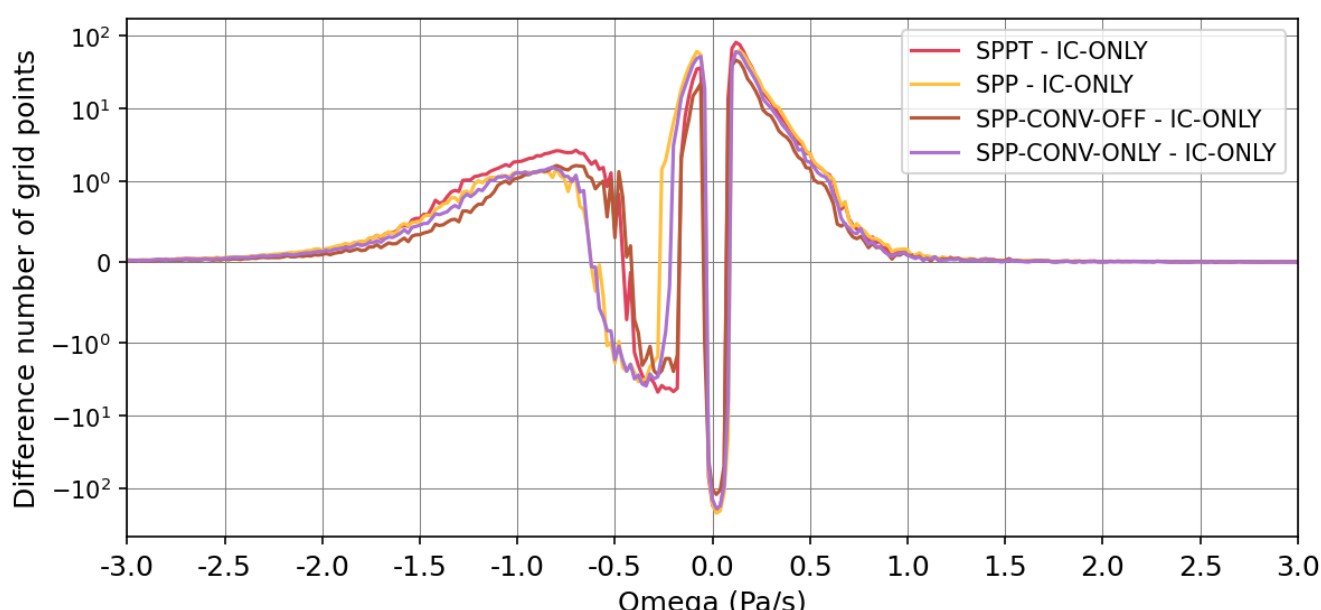

**Figure 4.** Difference number of grid points of vertical velocities at 500 hPa between the experiments SPPT (red), SPP (yellow), SPP-CONV-
OFF (brown), and SPP-CONV-ONLY (purple), and the experiment with unperturbed model physics IC-ONLY. Averaged over 11 forecasts.
The bin width is 0.02 Pa/s. Note that the y-axis has a linear scale between -1 and 1, and a logarithmic scale for values smaller than -1 and
larger than 1.





to the model physics (e.g. perturbations to the dynamical core). Comparing such schemes in this context could advance the understanding which properties of perturbation schemes result in a one-sided model response. This is, however, beyond the scope of this study.

## 3.2 Implications for the model climate

It was shown that stochastic perturbations in the forecast model have a systematic impact on rapidly ascending air streams and on vertical velocities in general. Up- and downward motions are an important component of the atmospheric circulation and are linked to atmospheric phenomena on different spatio-temporal scales. Therefore, imprints of the modulations of vertical velocities should also be reflected in simulated weather activity that is directly or indirectly linked to vertical motions. In this section, we evaluate the impact of stochastic model uncertainty schemes two such phenomena: precipitation (section 3.2.1) and the representation of the upper-level Rossby wave amplitude (section 3.2.2).

### 3.2.1 Precipitation

Following Pickl et al. (2022), who have investigated the impact of SPPT on global precipitation sums, we expand their analysis to SPP and its variants. In Figure 5, the differences of the number of grid points of precipitation rates between the experiments with perturbed forecast models and the unperturbed experiment are shown. As for the previously discussed diagnostics, the experiments with model physics perturbations (SPPT and SPP) show a very similar pattern with increased frequencies for

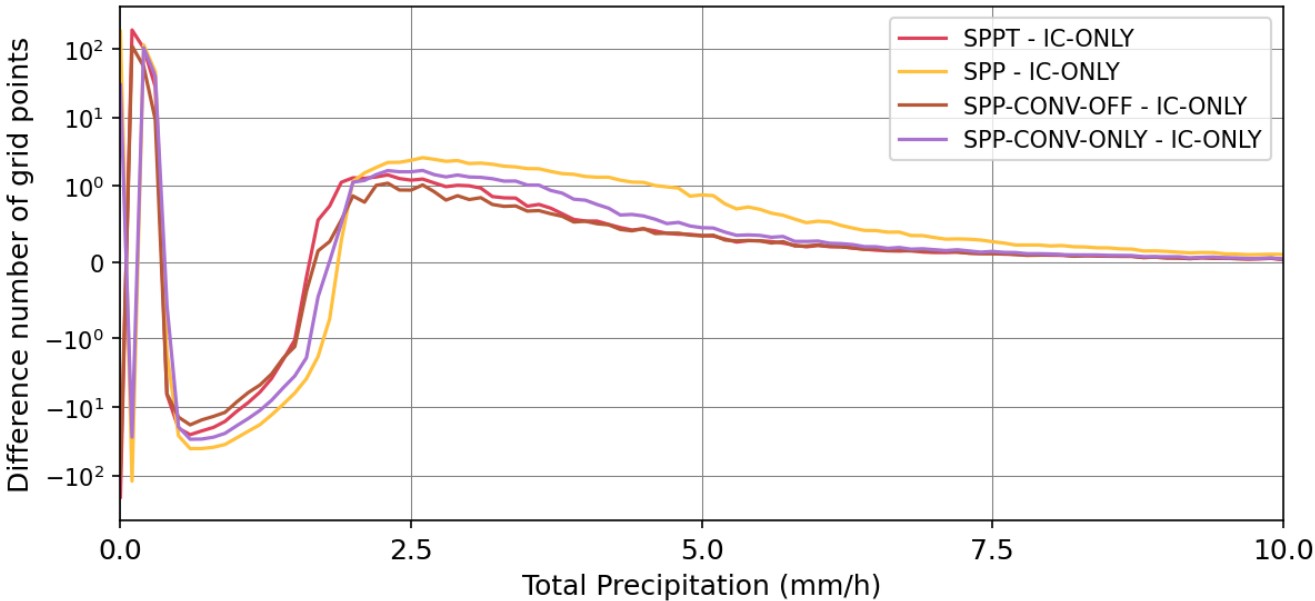

**Figure 5.** As Figure 4, but for precipitation rates (mm/h) and a bin width of 0.1 mm/h.





precipitation rates below 0.5 mm/h and above 1.8 mm/h for SPPT and 2 mm/h for SPP, whereas the number of grid points
with precipitation rates in between is decreased compared to the unperturbed experiment. The SPP sensitivity experiments
qualitatively show a similar behaviour as SPP and SPPT, but perturbations only in the convection scheme result in a larger
increase of the largest precipitation rates than perturbations to all other parametrisations. The patterns of all experiments
strongly resemble the bimodal structure of the changed mid-tropospheric upward motions (cf. Figure 4). Thus, the modulation
of upward motion goes along and might control the modulation of the precipitation frequencies.

### 295   3.2.2   Large-scale extratropical circulation

Due to the large impact that rapidly ascending air streams in the form of WCBs exert on the evolution of the large-scale flow
through ridge amplification (e.g. Grams et al., 2011), the question arises if the signal of the altered occurrence frequencies of
WCBs through stochastic perturbations is propagated upscale and whether the upper-level Rossby wave pattern is amplified
through model uncertainty schemes. We approach this question by comparing the sums of the areas of objectively detected
upper-level ridges and troughs (i.e. the Rossby wave amplitude, c.f. chapter 2) in the experiments with and without model
uncertainty representations. Due to the limited sample size of this diagnostic (i.e. only one value per forecast valid time), we
omit the experiments SPP-CONV-ONLY and SPP-CONV-OFF with only 11 initial times.

### 3.2.3   Equivalent latitude

The classification of each grid point required to determine the Rossby wave amplitude depends on the data set which is used
for the identification of the equivalent latitude $\Phi_{eq}$. Prior to the analysis of the Rossby wave amplitude, we therefore investigate
the behaviour of $\Phi_{eq}$ (i.e. the average latitude of the Rossby wave pattern) and its differences between the simulations. Figure 6
shows the mean evolution of $\Phi_{eq}$ in the different experiments (colored lines) as well as in the reanalysis dataset (grey line) with
forecast lead time on different isentropic levels. Note that the drift of $\Phi_{eq}$ in ERA5 results from the investigation period that lies
at the transition between late summer and autumn (c.f. section 2). The average $\Phi_{eq}$ on an isentropic surface follows the yearly
temperature cycle and is located further poleward in the warm season than in the cold season, resulting in a southward shift
of 2–3 ° during the considered 12 days of lead time (Figure 6). Note that the diurnal cycle present in both the reanalysis and
experiment datasets is a result of temperature-induced fluctuations of PV in regions of high topography (e.g. over the Tibetan
Plateau), where the isentropic surfaces intersect topography.


Comparing the experiment SPPT to IC-ONLY shows that, despite identical initial conditions, the southward drift of $\Phi_{eq}$ is
more pronounced in the experiment without model physics perturbations on all isentropes. This displacement of $\Phi_{eq}$ in SPPT
with respect to IC-ONLY corresponds to a pole- and upward shift of the hemispheric tropopause with stochastic perturbations.
Apart from 320 K, where the differences between the datasets are minor, $\Phi_{eq}$ decreases at a faster rate in the reanalysis data set
than in the experiments. The representation of $\Phi_{eq}$ in unperturbed experiment is therefore more consistent with reanalysis than



**Figure 6.** Evolution of $\Phi_{eq}$ with forecast lead time in the experiments and in ERA5 at the isentropic levels (a) 320 K, (b) 325 K and (c) 330 K. Averaged over 20 ensemble members and 32 initial times.



in the experiment with SPPT. The evolution of $\Phi_{eq}$ in SPP largely follows SPPT.

The differences between the experiments have important implications for the determination of the Rossby wave amplitude, as the computation of the ridge and trough areas are based on the chosen $\Phi_{eq}$. In our case, the choice of an independent $\Phi_{eq}$ (as done in Gray et al. (2014)) results in biased ridge and trough areas, as the Rossby wave patterns are shifted relative to each other, but are evaluated against an identical $\Phi_{eq}$, which does not represent the corresponding dynamically coherent, hemispheric-mean background state. Therefore, instead of using $\Phi_{eq}$ of a reference data set, we use $\Phi_{eq}$ of each individual experiment/reanalysis for each ensemble member and valid time to detect the upper-level troughs and ridges in its respective model background. This approach results in balanced areas of ridges and troughs on a given isentropic surface.

### 3.2.4 Rossby wave amplitude

Figure 7 shows how the area of upper-level ridges and troughs (i.e. Rossby wave amplitude) evolve on average with forecast lead time on different isentropic levels. The analysed amplitude (ERA5, grey line) on the 325 K isentrope (panel b) amounts to $2.35 - 2.4 \cdot 10^7 \, \text{km}^2$ , which corresponds to about 9% of the area of the Northern Hemisphere. On 320 K (panel a), the amplitude increases with "lead time", whereas it decreases on 330 K (panel c). This is again due to the experimentation period in the transition time from summer to autumn, when ridge and trough areas become larger on lower isentropes. Note that the cyclic behaviour, especially at 320 K is due to the short data period with forecast initialisations every second day, resulting in auto-correlated time series and a distinct impact of single events. Analysing the evolution of the Rossby wave amplitude in the experiment IC-ONLY (blue lines in Figure 7) clearly indicates that the forecasts underestimate the waviness of the upper-level flow on all isentropes. On 320 and 325 K, the amplitude is only slightly underestimated until forecast day 6–7 (144–168 h). Afterwards the difference to ERA5 becomes larger, ending up with a reduction of the amplitude of approximately $2.8 \cdot 10^6 \, \text{km}^2$ (11%) at 320 K at forecast day 12 compared to ERA5. On the other isentropic levels, the underestimation is not as pronounced as on 320 K, but still amounts to 8% on 325 K and 4% on 330 K with respect to ERA5. These results show that undulations of the upper-tropospheric wave guide become less pronounced with forecast lead time. This results in an overall too zonal flow configuration on the hemispheric scale. The model drift to less amplified Rossby waves is a well-known systematic bias of NWP models: Gray et al. (2014) report a systematic decrease of the ridge areas in the northern hemisphere in several winter seasons and in different models. They argue that erroneous representations of diabatic processes are a possible reason for these systematic errors, while Martínez-Alvarado et al. (2018) found that the decrease of the Rossby wave amplitude can partly be attributed to deficiencies in the dynamical core of forecast models. From a dynamical perspective this behaviour is related to a decrease/underestimation of the PV gradient along the dynamical tropopause (i.e. tropopause sharpness) with forecast lead time, which is another long-standing issue of NWP models (Gray et al., 2014; Saffin et al., 2017; Martínez-Alvarado et al., 2018; Schäfler et al., 2020). We have, however, omitted this aspect of Rossby wave dynamics in our study and focus on the amplitude only.



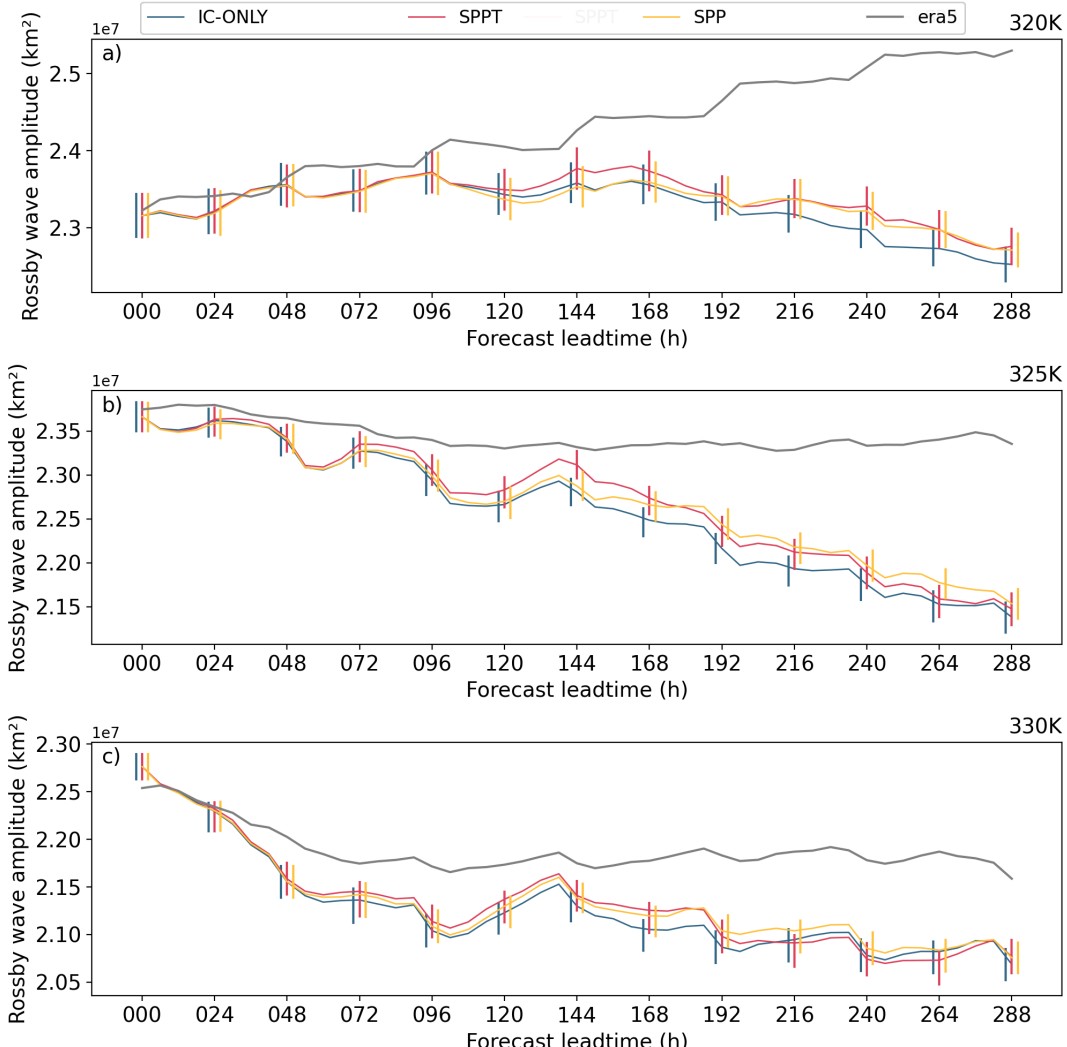

**Figure 7.** The lines show the mean evolution of the Rossby wave amplitude with forecast lead time in the experiments IC-ONLY (blue), SPPT (red) and SPP (yellow) and in ERA5 (grey) on the isentropic levels (a) 320 K, (b) 325 K and (c) 330 K. Averaged over all 32 forecasts and 20 members. The bars display the 10–90% confidence interval for each experiment computed by sub-sampling the dataset 10000 times (shown only every 24 h).

Next, we analyse in more detail the effect of stochastic parametrisations on the Rossby wave amplitude in the model forecasts. In the experiment with SPPT, the amplitude is increased compared to the experiment without model physics perturbations. At forecast initialisation, SPPT and IC-ONLY have identical values (both start at the same initial conditions), but the areas gradually decrease with forecast lead time in IC-ONLY compared to SPPT. This is especially visible at 320 K, where the amplitude on forecast day 12 in SPPT is increased by approximately 1% (equivalent to an area of $2.3 \cdot 10^5 \, \mathrm{km}^2$) compared to



IC-ONLY. On 325 K, this behaviour is also apparent, but less pronounced. On 330 K, the Rossby wave amplitude in SPPT is increased up to forecast day 8 (192 h), but is then very similar or even larger in IC-ONLY until the end of the forecast. Still, the general picture is that SPPT slightly increases the amplitude of the upper-level Rossby wave pattern compared to unperturbed physics, especially on 320 and 325 K. As the experiments generally underestimate the Rossby wave amplitude compared to reanalysis, the increase by SPPT results in an improvement of the representation of the waviness of the upper-level circulation.

SPPT thereby helps to maintain the upper-level ridge and trough areas against the systematic lead-time dependent degeneration of the Rossby wave amplitude. The magnitude of the effect is, however, quite small, and the confidence intervals of the experiments mostly overlap.

Similarly to the SPPT-scheme, also SPP increases the amplitude of the upper-level Rossby wave pattern compared to the

unperturbed reference simulation: especially on 320 and 325 K, the amplitude is larger with SPP than without model perturbations. Compared to SPPT, the effect is slightly smaller, and the lines of IC-ONLY and SPP diverge at later lead times than IC-ONLY and SPPT.

Overall, we observe differences between the experiments with and without stochastic parametrisations, and the increased

Rossby wave amplitude is consistent with the increased frequency of rapidly ascending air streams in the Extratropics. Nevertheless, the signals are rather small and the differences lie within the confidence intervals of the experiments. To increase the confidence in our results, we analyse the Rossby wave amplitude in an additional data set in the subsequent section.

### 3.2.5  Inferences from operational (re-) forecasts

Due to the experimental setup consisting of only 32 forecasts and 20 ensemble members (i.e. 640 cases), and the fact that the

ridge area diagnostic only yields one scalar number per valid time, the robustness of the previously discussed results of the lead-time dependent evolution of the Rossby wave amplitude has to be demonstrated. We attempt to do this by making use of the findings from Pickl et al. (2022) and substantially enlarge the sample size by comparing the Rossby wave amplitudes in the unperturbed control member of operational ECMWF reforecasts to the perturbed members (SPPT + ICP). We analyse in total 3,200 ensemble forecasts initialised in winter, spring and autumn between 1997 and 2017, which, additionally to increasing

the sample size, allows us to investigate seasonal differences of the effect of stochastic parametrisations on the Rossby wave structure. Note that the perturbed forecasts have 10 members, while the unperturbed control member is a deterministic run, which results in different sample sizes of the two data sets (n=32,000 for the perturbed and 3.200 for the unperturbed forecasts).

Figure 8 shows the mean evolution of the Rossby wave amplitude of the perturbed (solid) and unperturbed (dashed) mem-

bers as a function of lead time for forecasts initialised in winter (a), spring (b) and autumn (c; note the different ranges of the y-axes). In all seasons, the Rossby wave amplitude is on average larger in the perturbed forecasts than in the unperturbed forecasts, with the largest signal in autumn (on average 0.65% difference), followed by winter (0.45%) and spring (0.2%). In winter (panel a), the differences between perturbed and unperturbed members are largest until lead times of about 4–7 days




(about 1% difference). After that, the difference between the two data sets does not increase and even vanishes around forecast
day 15. Perturbed forecasts initialised in spring (panel b) are characterised by only slightly increased Rossby wave amplitude
compared to the unperturbed forecasts (maximum of about 0.6–0.8% at day 11, but mostly smaller values are present). In au-
tumn, the difference of the Rossby wave amplitude between perturbed and unperturbed members is largest and increases with
forecasts leadtime, reaching differences of up to 0.8–1.2% during lead times of 12–15 days. Surprisingly, the largest difference

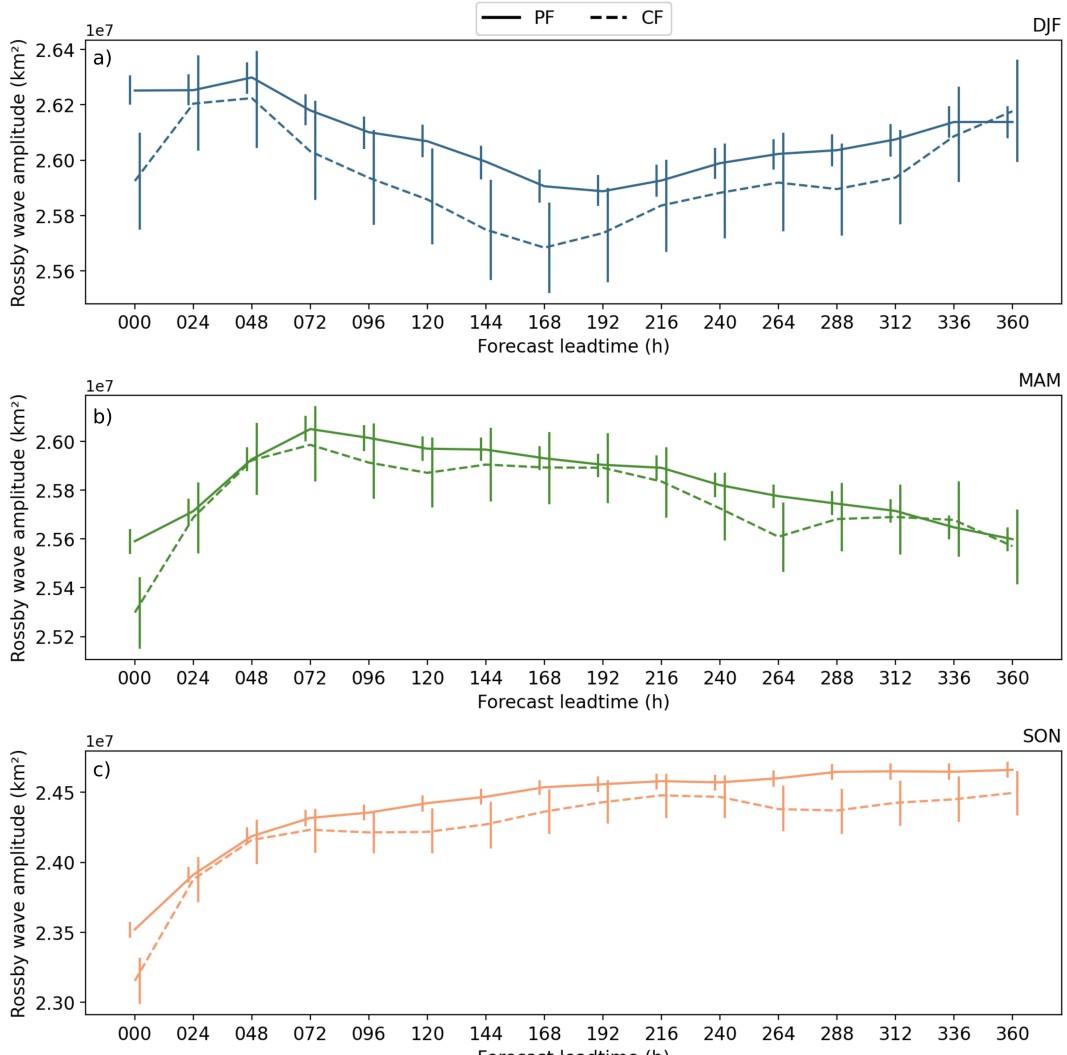

**Figure 8.** Average evolution the Rossby wave amplitude at 320 K with forecast lead time for perturbed (PF, solid lines) and unperturbed
reforecasts (CF, dashed lines) initialised twice weekly between 1997 and 2017 in winter (a, blue), spring (b, green) and autumn (c, orange).
The bars display the 10–90% confidence interval for the two data sets at each valid time, computed by resampling the dataset 5000 times.
Note the different ranges of the y-axis in each panel.





in all seasons is observed for the initial time. This shows that the initial condition perturbations strongly increase the waviness
of the upper-level flow for the first few forecast hours.

The comparison of perturbed and unperturbed members of ECMWF reforecasts confirms the main findings from the sensitivity experiments in the previous section. Even though only one isentropic level ($\theta$=320 K) and one scheme (SPPT) is investigated, the larger sample size allows for an assessment of the robustness of the observed patterns. In the experimentation
period, which is mainly in autumn 2016, the amplitude of the upper-level wave pattern on 320 and 325 K is increased when perturbations through SPPT/SPP are active. In this analysis, a quantitatively very similar pattern is found for the reforecasts initialized in autumn, where the order of magnitude ($\mathcal{O}(1\%)$) of the observed effect is similar. This behaviour can therefore be considered as robust, even though the sampling variability is mostly larger than the signal (see overlapping bars in Figures 7 and 8). This uncertainty could most likely be reduced by further increasing the sample size, e.g. with a larger ensemble or more
initial dates. On the other isentropic levels in the experiment data set as well as in other seasons than autumn in the reforecast data set, the signal is not as distinct as in autumn and on 320 K.

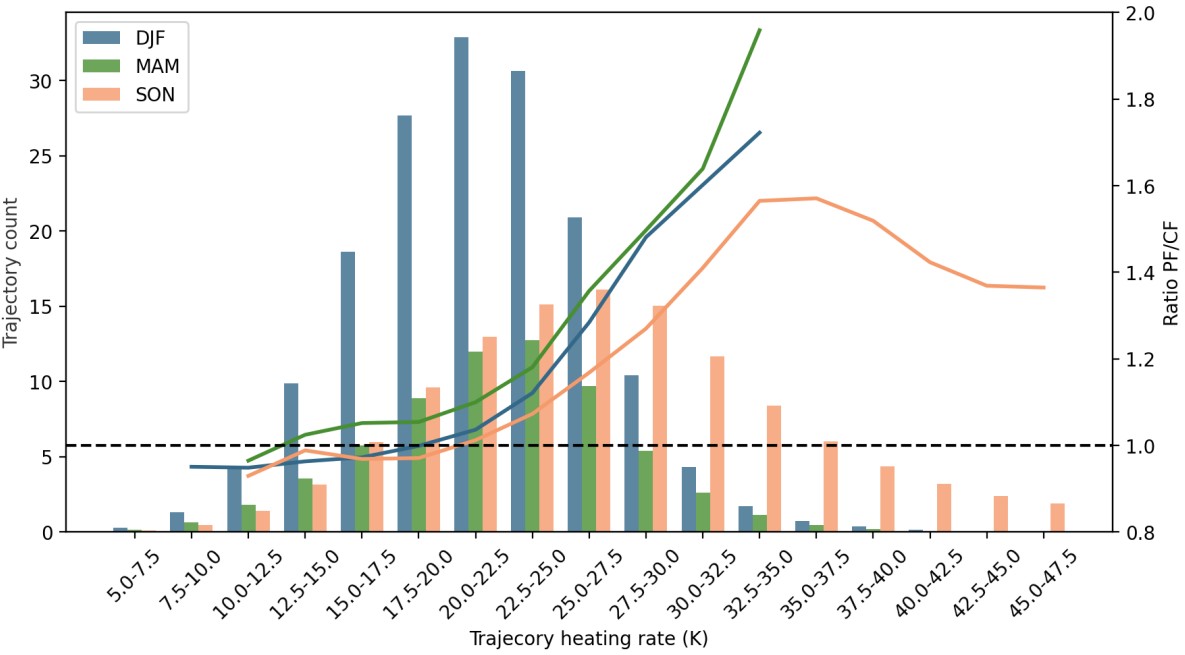

**Figure 9.** WCB trajectory counts in perturbed members (left axis, colored bars) and ratio of WCB trajectory numbers between operational medium-range forecasts forecasts with and without perturbations (right axis, colored lines) initialized in winter (blue), spring (green) and autumn (orange) for operational ECMWF medium-range forecasts during the period Dec 2018 to Nov 2021.





Due to the coarse resolution of the archived reforecast data set, it is not possible to detect ascending air streams using tra-
jectory analysis. We therefore make use of another data set that consists of 2 years of operational medium-range ensemble
forecasts archived regionally for the North Atlantic domain, in which trajectory data is available (c.f. Section 2.4). Thereby,
we assess whether the inter-seasonal differences in the modulation of the amplitude of the upper-level flow through SPPT
(Figure 8) are in agreement with the impact of the model uncertainty schemes on rapidly ascending air streams. By comparing
WCB trajectories in the unperturbed and perturbed members, the seasonal differences of the effect SPPT on the trajectory
counts can be evaluated. Similarly to Figure 3, Figure 9 shows that the ratio of the number of rapidly ascending trajectories in
the perturbed forecasts and in the unperturbed forecasts is largely controlled by the latent heating rate along the ascent. The
inter-seasonal differences are in agreement with what has been observed for the seasonal differences of the impact of SPPT on
the Rossby wave amplitude: in autumn, the latent heating rates along rapidly ascending air streams are largest and therefore
their frequency is increased the most through stochastic perturbations, and also the Rossby waves are affected most strongly. In
winter, the effect on the rapidly ascending air streams is smaller than in autumn, as the latent heating rates are decreased, and
consequently, also the effect on the Rossby wave amplitude is smaller. In spring, the latent heating rates are comparable to those
in winter, but the absolute frequency of WCBs is much smaller and the diabatic influence on upper-level ridge building is not as
pronounced as in winter and autumn, which is consistent with the weakest impact on the Rossby wave amplitude in spring. This
joint sensitivity of the seasonal modulation of the rapidly ascending trajectories and the Rossby wave amplitude to SPPT is a
strong indication that the two signals are linked to each other. This insight holds even though two different data sets have been
used, as the impact of SPPT on the occurrence frequency of ascending trajectories has been shown to be robust for different se-
tups of the IFS (Pickl et al., 2022); therefore, we assume that the WCB-modulation through SPPT is similar in the two data sets.

## 4 Discussion

In this study, we investigate the influence of different model perturbation techniques in ECMWF's ensemble prediction system
on rapidly ascending, diabatically driven air streams, and explore the impact of these sensitivities for the representation of
weather phenomena that are related to ascending motions. This section discusses the presented results and aims to link the
findings of the individual sections in order to develop a process-oriented understanding of how the stochastic perturbations
result in the observed responses.

SPPT and SPP, which both represent model uncertainty related to physical parametrisations, have a very similar impact on
rapidly ascending air streams and systematically increase their occurrence frequencies. The impact of SPP is more pronounced
in the tropics, whereas SPPT has a larger impact in the extratropical regions. This is in agreement with Leutbecher et al.
(2017), who report similar regional differences for ensemble spread between SPPT and SPP. Similar sensitivities are evident
when only parameters in specific parametrisations are perturbed, as shown by experiments with perturbations only in the con-
vection parametrisation and in all parametrisations but convection. This indicates that the unilateral response of the model does





not depend on how the perturbation is introduced (i.e. perturbation of the net tendencies or perturbation of the parameters), but where it is applied and if the perturbation is taken up by nonlinear processes. The local amplitude of the perturbations, nevertheless, is a crucial factor in triggering a one-sided response in the system. For both SPPT and SPP, the amplitude of the perturbations is on average large in regions of rapidly ascending air streams, as these are typically associated with large
parametrisation tendencies (e.g. by the convection and microphysics schemes).

To determine whether the altered distributions of rapidly ascending motions have an impact on other processes in the forecast model, the response of precipitation and the large-scale flow to the different model uncertainty schemes has been analyzed. The impact of the stochastic perturbation schemes on precipitation can be directly attributed to the altered vertical velocities,
as the bimodal structure of the modifications of the upward motions with stochastic parametrisations is clearly reflected in the changes to precipitation.

The effect of the perturbations on the upper-level Rossby wave amplitude, measured by the hemispheric sum of the areas of upper-level ridges and troughs, and their connection to the altered occurrence frequencies of WCBs, is more complex. The
initial hypothesis that the increased diabatic outflow from WCBs through stochastic model uncertainty schemes should result in a more amplified upper-level flow can be confirmed insofar that the ridge and trough areas are larger with than without model perturbations. In our study, we were able to show this by means of sensitivity experiments and by analysing operational reforecasts. The order of magnitude of the effect of SPPT and SPP on the rapidly ascending trajectories is, however, one order of magnitude larger than the effect on the upper-level troughs and ridges: with SPPT, for example, the counts of trajectories
that are detected as WCBs are increased by approximately 10-20% in the northern hemisphere Extratropics (exact numbers depend on the season, c.f. Figures 2 and 9), while the ridge and trough areas are increased by only up to 1%. Comparing these two numbers with each other, however, has to be done with caution, because of the following considerations:

  – The divergent outflow of diabatically enhanced ascents is not the only process which contributes to the formation and amplification of upper-level ridges. For example, Teubler and Riemer (2021) highlight that barotropic wave dynamics
at the tropopause, the interaction of baroclinic cyclones with the upper-level flow, and other diabatic processes (such as radiation and turbulence) influence troughs and ridges. Without the knowledge of how these processes are changed by stochastic perturbations, it is not possible make a quantitative statement on the role of altered WCB frequencies for the ridge and trough areas. To the authors' knowledge, however, there are no studies on systematic effects of perturbations on these processes.

– The trajectory count relies on the choice of a threshold (ascent of at least 600 hPa within 2 days). Air streams that fail to fulfill this criterion (maybe by only a few hPa) are not considered in this diagnostic, even though they might exert a similar impact on the upper levels. The additional diabatically induced divergent outflow at the tropopause level responsible for the ridge amplification is therefore likely not of the same order of magnitude as the trajectory count.



    – The dynamical response of the upper-level jet to the divergent outflow of WCBs depends on the distance of the outflow
to the jet and on the outflow height relative to the tropopause level (Grams and Archambault, 2016). The tropopause
      height and the latitude of the 2 PVU line (i.e. $\Phi_{eq}$) are both increased through the perturbations (c.f. Figure 6), which
      makes it more "difficult" for the WCB outflow to impinge on the jet stream and to initiate or amplify ridge building,
      assuming similar characteristics of the WCB outflow (e.g. outflow height and latitude) with and without perturbations.

From these aspects, it becomes clear that the modulated WCB-frequencies do not directly project to the changes of the
Rossby wave amplitude. Nevertheless, a causal relationship between the effect of stochastic perturbation schemes on the
WCB-occurrence and on the upper-level Rossby wave patterns is supported by the following considerations:

    – While stochastic perturbations affect the vertical velocities in the forecasts immediately after they have been applied
      (Pickl et al., 2022), the Rossby wave amplitude with and without model perturbations is identical at the beginning of
      the forecasts and diverges very slowly with forecast lead time. This points towards a weak, yet constant forcing of the
increased mass flux through the enhanced WCB activity on the upper-level ridges. The saturation of the process (i.e. the
      decreasing and vanishing differences of ridge and trough areas between perturbed and unperturbed forecasts, c.f. Figures
      7 and 8) could arise from compensating effects of the model, without which the differences would constantly grow with
      forecast lead time.

    – SPP alters the vertical velocities and the frequency of WCB-trajectories in a very similar way as SPPT, even though the
perturbation techniques differ from each other.

    – The analysis of the (re-) forecast archives have shown that the trajectory counts and the Rossby wave amplitude are
      modulated similarly across the seasons, with the strongest impact in autumn and a weaker signal in winter and spring.
      This points towards a direct interrelation and a common underlying mechanism of the processes.

    Our investigation adopts a novel, process-based perspective on the effects of stochastic perturbations on the large-scale ex-
tratropical circulation. In previous studies, it has been shown that stochastic model perturbations improve the representation of
Euro-Atlantic weather regimes across different model hierarchies (Dawson and Palmer, 2015; Christensen et al., 2015), espe-
cially for such regimes that are characterized by blocking anticyclones. In agreement with the presented results, the reported
impacts on the large-scale circulation are mostly very subtle, especially in numerical models of high complexity (e.g. Davini
et al. 2021, Dorrington 2021). Christensen et al. (2015) argue that stochastic forcing enables a more realistic sampling of
Lorenz-like attractors in models of reduced complexity, as the introduced noise helps to transition between stable states of the
system (i.e. noise-induced regime transitions; Berner et al. (2015)). Dorrington (2021) mention that improved representations
of the Atlantic ridge regime in fully coupled simulations with SPPT might be driven by improved tropical modes of variability
(i.e. ENSO) whose signal is transferred to the Extratropics via teleconnections. Martínez-Alvarado et al. (2018) state that dif-
ferences in the sharpness of the tropospheric wave guide between perturbed and unperturbed forecasts are directly induced by
vorticity perturbations along the large gradients at the dynamical tropopause. With our analysis, we contribute to the discussion
and propose a coherent process chain how the random perturbations affect the model climate: the distinct threshold behaviour

in the dynamics of rapidly ascending, diabatically enhanced air streams results in a one-sided response of the symmetric per-turbations through SPPT or SPP. The increased occurrence frequency of the ascending air streams, for example in the form of WCBs, is then projected to related weather phenomena, such as the upper-level Rossby wave pattern, which ultimately changes
the model climate of the large-scale circulation.

## 5   Conclusions and Outlook

With the presented analysis, we compare the effect of different model uncertainty schemes applied to ECMWF's ensemble prediction system on rapidly ascending air streams and provide a coherent explanation how stochastic perturbations can in-fluence the mean state of the forecast model. We thereby contribute to the discussion how the large-scale extratropical flow
is modified through stochastic model perturbations on a process level. We argue that stochastic perturbation schemes change the distribution of precipitation and amplify the upper-level Rossby wave pattern by modulating the occurrence frequency of vertical motions (especially of rapidly ascending, moist air streams), which occurs due to the nonlinear nature of systems that are characterized by threshold behaviour (Pickl et al., 2022).

In order to further substantiate the causal relationship between the modulated ascending air streams and the increased Rossby wave amplitude further sensitivity experiments have to be conducted. Apart from more ensemble members or forecast initiali-sations, which could yield a more robust signal, an experimental setup with perturbations confined to specific height layers of the troposphere (e.g. only in the upper troposphere) would shed light on whether the observed effect is caused locally or by a vertical propagation of the perturbations. Similarly, perturbations could be applied solely in pre-defined regions (e.g. only in the
tropics) to determine if the signal originates from remote regions and is propagated by tropical-extratropical teleconnections. Further, a more quantitative framework that uses PV-tendencies to evaluate barotropic, baroclinic and divergent contributions to upper-level flow features (Teubler and Riemer, 2021) could be applied to ensemble experiments to gain further insights how perturbations affect the model dynamics. However, all of these proposed approaches require a significant amount of resources in regards of experimental design / model implementations and computational power, and are therefore beyond the scope of
this study.

*Data availability.* The S2S reforecast data set is available publicly from https://apps.ecmwf.int/datasets/data/s2s/levtype=sfc/type=cf/, and the ERA5 reanalysis data set can be downloaded from https://doi.org/10.24381/cds.bd0915c6 (Hersbach et al., 2020). The experiment data and the operational medium-range forecast data set are too large to upload and will be shared upon request.

*Author contributions.* MD conducted the numerical IFS experiments, performed all analyses presented in the paper and wrote the manuscript. CMG downloaded the operational ECMWF forecasts and gave important guidance and feedback during all stages of the project.



*Competing interests.* One of the authors is a member of the editorial board of Weather and Climate Dynamics.

*Acknowledgements.* This work was funded by the Helmholtz Association as part of the Young Investigator Group "Sub-seasonal Predictabil-
ity: Understanding the Role of Diabatic Outflow" (SPREADOUT, grant VH-NG-1243). ECMWF and Deutscher Wetterdienst are acknowl-
edged for granting access to computing facilities and real-time operational ensemble forecast data. We are very grateful to Simon Lang from
ECMWF, who significantly contributed to the design of the numerical IFS experiments and helped with the interpretation of the results. We
thank Heini Wernli and Michael Sprenger for providing the LAGRANTO software. We are grateful to Martin Leutbecher and Sarah-Jane
Lock from ECMWF and to the members of the Large-Scale Dynamics and Predictability group at KIT for valuable discussions on this
project.



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
