# Peer review of "Towards a process-oriented understanding of the impact of stochastic perturbations on the model climate"

_EGUsphere, 2023_

## Author Response (AR1)

**Response to reviewer comments for "Towards a process-oriented understanding of the impact of stochastic perturbations on the model climate"**

**Initial submission on Aug 24, 2023, EGUSPHERE-2023-1938**

Deinhard, M. and Grams, C.M.

Submission of revised manuscript on March 3, 2024

Dear Mr Pedram Hassanzadeh, dear reviewers,

thank you for the constructive feedback on our paper. We have prepared a revised version of the document, where we took into account the comments from the reviewers. Point-by-point answers are given (in blue) in this document.

**Reviewer 1**

**General comments**

The paper presents a study on how model error representation in the operational NWP forecast model of ECMWF can affect the model climate state. Here, stochastic perturbations to either the net physical tendencies or physical parameters for the parametrisation schemes are used to account for errors in the unresolved physical processes of the model. Their impact on the occurrence of diabatically driven, rapid ascending air streams – using trajectories – is analysed, resulting in systematically more frequent situations with such rapidly ascending air flows compared to unperturbed simulations without model error representations. The two considered stochastic schemes produce broadly similar results. Interestingly, the one-sided response to the stochastic forcing cannot be attributed to a single parametrized process (convection). It was further demonstrated how these systematic effects are directly linked to global precipitation statistics and to the amplitude of upper-level Rossby wave patterns. It was found that both stochastic schemes increase the waviness of the upper-level flow and thereby reduce the systematic bias of the model, even though the magnitude of the effect is small.

I really enjoyed reading this paper and think it is great addition to the existing literature on the effects of stochastic physical perturbations for model error contributions in numerical models. In particular, the process-based approach to understand the impact of the perturbations from the latent heat release along the ascending air streams to vertical velocity and precipitation and subsequently on the large-scale Rossby waves (amplitudes) is a very welcome advance over the often more statistical-in-nature studies that were carried out in the past. The experiments are well motivated, the results are both very interesting and presented in a clear and compelling way. I congratulate the authors on a great paper.

I only have a few minor suggestions and would certainly suggest publication of this study.

Thank you very much for your positive feedback on our study!

**Detailed comments**

- Line 44: it might be worth to emphasise here that the SPPT scheme applies multiplicative random noise with certain spatial and temporal autocorrelation scales.
Thank you for this remark. As this is a more general introduction about stochastic parametrisations, we prefer to include the suggested additions in the methods section (section 2.1). There, we have changed the corresponding sentence which now reads (lines 113–114): "It perturbs the model physics by multiplying the net tendencies from all parametrisations with a random field which evolves in space and time with prescribed autocorrelation scales."

- Line 125: specify the year for which the experiments were run
Thank you for spotting this. We have included the year of the experiments.

- Section 3.1: discuss how different the various experiments perform with regards to the verification (ANA), mention in the discussion which of the differences are significant (the confidence intervals are not mentioned in the text even though they are plotted), in particular for the non-significant differences
Thank you for this comment. We have now included a discussion about how the experiments perform compared to the verifying analysis (see lines 216 ff.) Further, we have changed Figure 2 such that the whiskers show the 95% confidence intervals of the mean. In the previous version, the whiskers were a measure of variability across the dataset (i.e. standard deviation across forecasts, lead times and members). The confidence intervals are very narrow and almost all differences are significant. We have included a comment on the bootstrapping in the caption to Figure 2.

- Same section and Fig 3: it would be helpful to briefly offer an interpretation of the heating rates. Maybe label the x-axis in the figure as heating rates in K/6h.
Thank you for this comment. We have rewritten the discussion on Figure 3 and given some more interpretation on the heating rates (lines 221 ff.), including a reference to Madonna et al. (2014). We stick to the notation of net heating during the ascent period (i.e. K/48 hours), as this is frequently used in the literature and as it is directly linked to the ascent time scale.

- Section 3.2.1: Does the range of negative differences in Fig 5 coincide with the range of the drizzle overestimation problem in many NWP forecast models?
Thank you for this interesting question. According to the AMS glossary, drizzle results in precipitation rates of max. $0.5\,\mathrm{mm/h}$ (`https://glossary.ametsoc.org/wiki/Drizzle`). However, in our study the differences between the experiments and the reference are positive up to this threshold. This would suggest that the perturbations result in an even larger overestimation of drizzle.

- Fig 6: could confidence intervals be added, similar to Fig 7?
Thank you for this remark. We have added confidence intervals to Figure 6. Additionally, we also have included information on statistical significance in most other Figures (all except for Figures 4 and 5).

**Reviewer 2**

**General comments**

This manuscript explores how stochastic model uncertainty perturbations impact rapidly ascending airstreams, also precipitation patterns and upper-level flow, providing a convincing process-level description of mean state changes that can result from the stochastic schemes. This work builds on previous work by the authors, which explored the impact of SPPT perturbations on WCBs.

The work is well designed and conducted. The presentation is very good and it was an enjoyable read. I think the study will make a valuable contribution to the literature. I have some suggestions for minor improvements – some specific and some more broad. None require much additional work, but they would improve the quality of the presentation.

One general criticism (detailed examples are described below): there are a number of instances in the manuscript where the authors make statements that imply greater breadth in the work than is presented or appear to overstate the size of differences in the experiment results. The work is very interesting, well-designed and well presented. There is no need to overstate the claims. Indeed, exposing the limitations highlights the areas that would benefit from further studies. I would ask the authors to be careful to present the work with complete accuracy – rely on the quality of the work to expose its merits and not consciously overstate them.

Thank you very much for your positive feedback on our study. We acknowledge your criticism that at some points of the paper, statements appear to be too general and differences between experiments are very small. However, we did not intent at all to overstate our results. We are well aware that some of the signals, especially the ones regarding differences between experiments regarding precipitation and the Rossby wave amplitude, are subtle, and admit that some of our formulations are not as precise as they should be. We therefore take your feedback very seriously and have considered most of your suggestions (see below for detailed answers to all points). In summary, we have implemented the following changes into the revised manuscript:

- We have added measures of statistical significance to several of our results based on a bootstrapping approach (Figures 2, 3, 6, 8 and 9).

- We have reformulated sentences in which we have discussed small differences between experiments in detail and better focus on the message we want to convey.

- We carefully reviewed our statements regarding the implications of the comparison between the experiments SPP-CONV-ONLY and SPP-CONV-OFF.

**Detailed comments**

- Abstract (& throughout): results are described as showing that "perturbations to *different parameterisations* have similar effects". This a broad statement, which I find misleading. It implies more than what is shown in the paper. The SPP experiments demonstrate the impact of perturbations to the convection parameters alone (SPP-CONV-ONLY); and from perturbations to all other parameters (SPP-CONV-OFF). There are several parametrisations represented by the *CONV-OFF experiment, which could be interesting to explore, each in isolation (for a future study). I propose the authors take a less broad tone in describing the extent of the SPP exploration in this manuscript.
  Thank you for this comment. We agree with you that our conclusions from the comparison

of the experiments SPP-CONV-ONLY and SPP-CONV-OFF might be too general. As the project has expired, it is unfortunately not possible for us to run further experiments with other configurations of SPP, which would be needed to draw more general conclusions about 'different parametrizations'. We have rephrased the corresponding sentence in the abstract, which now reads: "The one-sided response to the stochastic forcing is also observed when only specific parametrizations are perturbed (only convection parametrization and all parametrizations but convection), and we hypothesize that the effect cannot be attributed to a single process.". In the discussion, we have included a sentence where we state that additional experimentation is required to for our hypothesis (lines 460 ff.): "To demonstrate this beyond doubt, it would be necessary to conduct additional sensitivity experiments with other configurations of SPP (e.g. with perturbations only to the boundary layer scheme), which is beyond the capabilities of this study". In Addition, we have reformulated alike statements in the manuscript in a similar way (see corresponding detailed comments).

- Related (e.g. line 194): the authors claim to analyse "other model uncertainty schemes [to SPPT]". Again, I find this misleading. The study explores one other MU scheme (SPP) but in several configurations. A more accurate description would be that "other model uncertainty *representations* [have been analysed]".
  Thank you for this comment. We have changed the text according to your suggestion.

- Line 24: remove "order of" – simply "the magnitude is small"
  We have removed "order of".

- Lines 55-73: to add to the discussion, a recent paper demonstrates that SPPT perturbations applied to an active MJO region can be used to explore and understand the pathways of error growth from the tropics to the extra-tropics. Straus et al. (2023), https://doi.org/10.5194/wcd-4-1001-2023
  Thank you for this reference. We added the citation (Straus et al., 2023) and a sentence along the lines of your suggestion (lines 59 ff.)

- Figure 1: the colour choice could be improved – the red and green can be difficult to distinguish. It is difficult to distinguish the blue and back lines (though the meaning is clear). In the shaded areas, there appear to be some red marks north of the equivalent latitude and some green marks to the south. According to the definition of troughs/ridges, this shouldn't be possible – is it an error in the plotting?
  Thank you for this feedback. We agree that the color choice is not optimal and therefore have changed the color scheme of Figure 1. Further, we have also corrected for the plotting error that you have spotted.

- Figure 2 and discussion (lines 195-212, also line 442): the differences between experiments do not all appear to be statistically significant: for the n. hem extra-tropics and the n. Atlantic, it is not obvious that there is any statistical significance in the differences between any of the experiments or the analysis. Unless I miss something, I would certainly refrain from making claims of differences between the SPP* experiments. Differences between others are perhaps "indicated"? For the tropics and globally, DET and IC-ONLY appear to be significantly different from the others; but, the error bars for the SPPT and SPP* experiments encompass the median of each of the others. If the differences are known to be statistically significant, please make that clear. If they are (known to be) not, please don't overstate the differences.

Thanks for the thorough analysis of the discussion regarding Figure 2. The whiskers in the original Figure are not a measure for sampling uncertainty, but display the variability across the ensemble members and time. In the revised version, we have included confidence intervals based on the same bootstrapping technique as in Figure 7, but at a higher level of significance ($\alpha$=0.05). Due to the large sample size, the differences are mostly statistically significant.

- Figure 3 and discussion (lines 220-230): further to the comment on Figure 2, the ratios of trajectory counts for the "extra-tropical regime" are very similar to each other. Without significance testing, I would be cautious about claiming (or believing there to be) any differences between the 4 experiments. For heating >40K, the differences do look clear and perhaps can be used to justify comments about differences between SPP, SPP-CONV-ONLY and SPP-CONV-OFF. Likewise, the comment (line 229) about SPPT for the smallest heating rates, given the small number of trajectories, I wonder whether the trajectory count ratio is really statistically different to 1.0?
  Similar to Figure 2, we have included a significance testing and show ratios in thick lines when they are significantly different from 1.0, which is the case for most data points (same for Figure 8). It is difficult to visualize whether the experiments are significantly different from each other, without losing the possibility to interpret the figure as a whole. Nevertheless, the fact that ratios close to 1 are significantly different from 1 gives an indication that also small differences between experiments can be interpreted. Still, we have changed one formulation in the corresponding section, which now reads (line 236): "[...] where the curve of SPP-CONV-ONLY is slightly higher than or similar to that of SPP."

- Figure 3 caption: mentions an experiment "STOCDP" and a lightblue line that is not present in the figure.
  Thanks for spotting this remnant from an older version of the paper. We have deleted the word.

- Line 240: the inequalities are incorrectly expressed: the maximum for SPPT for slow ascents occurs for -0.2 < $\omega$ < -0.05, and similarly for SPP.
  Thank you for spotting this, we have corrected the arrangement of the inequalities.

- Line 242: the omega range values are quoted the wrong way around (and incorrectly) for SPP and SPPT (according the figure): SPPT has a minimum for -0.4 < $\omega$ < -0.2, and similarly for SPP.
  Thank you for spotting this and apologies for the inaccuracy. We have corrected the values and the arrangement of the inequalities.

- Line 247: again, it is difficult to believe by eye from Figure 4 that the differences in the experiment lines for large +ve $\omega$ demonstrate real differences, without some indication of significance testing.
  It is indeed true that the differences are very small. We have included some expressions to emphasize that the differences between the experiments are small (e.g. deleted "qualitatively" in the sentence "model physics perturbations through SPPT and SPP result in qualitatively very similar changes of the occurrence of vertical velocities", wrote "slightly weaker" instead of "weaker" and "some minor differences" instead of "some differences"). Further, we have included an additional sentence (lines 270–271): "Nevertheless, the differences across the experiments with perturbed model physics are very small, indicating that the different perturbation techniques result in very similar changes to the vertical velocities.". With this, we emphasize once more that the behaviour between the experiments

is very similar.

- Line 252: "balanced": have you confirmed that the increased upward and downward mass fluxes generated by the stochastic perturbations do indeed balance?
No, we have not tested if the increased upward motion balances the increased downward motions. This should be understood as qualitative statement. We have included the word "qualitatively" at the beginning of the sentence and changed "balanced" to "compensated" (line 263).

- Line 258: "number of grid points [with what?] is decreased"
We have changed this sentence, which now reads (lines 268 ff.): "[...] shift of the velocity range in which the differences between the experiments with perturbed and unperturbed physics are negative."

- Line 263-266: to be clear: the "uni-directional response" being that the perturbations tend to result in more grid-points with non-zero vertical motion? Could the authors spell this out for the reader in the text.
Thanks for this comment. It's not only that the number of grid points with non-zero vertical velocity is increased, but also that very fast ascents occur more often at the expense of moderate ascents. We have included the following in brackets (line 277): "(i.e. acceleration of vertical velocities in two regimes)".

- Line 282, missing word: "uncertainty schemes *on* two such phenomena"
Thank you for spotting this, we have included the missing word.

- Line 287 & 290 & 320 (+ elsewhere?): not the "unperturbed experiment" but the "unperturbed *physics* experiment" or simply "IC-ONLY" (which includes initial perturbations)
We have changed this at all instances.

- Line 268: add a word for clarity: "increased *occurrence* frequencies" (to avoid confusion with precipitation frequencies)
We have included the missing word to avoid confusion.

- Line 294, missing word: "goes along *with* and might..."
Thank you for highlighting this. We have changed the sentence, which now reads (lines 306–307): "Thus, the modulation of upward motion is consistent with and might control the modulation of the precipitation frequencies".

- Figure 5: it is not easy to read from the image, but is there something interesting happening to grid-points with zero precip? Would it also be informative (even possible?) to indicate the number of grid-points (in IC-ONLY) for each precip rate (and omega in Fig 4)? Similar to what has been done in Figure 3. To give an impression of how widespread any changes in the rates are across the model.
Thank you for this comment. Indeed, the zero-precipitation data point behaves a bit odd: SPPT and SPP-CONV-OFF have negative differences, while SPP and SPP-CONV-ONLY have positive differences to IC-ONLY. However, we do not have a plausible explanation for this behaviour and prefer not to speculate. Regarding the suggestion of adding the total number of grid points for each precipitation rate: We have done this in a previous paper (in which we focused on SPPT) for the distribution of omega values (c.f. Pickl et al. (2022) Figure 7a, and attached Figure R1). By far largest fraction of grid points has values of very low vertical velocity, and the same is likely to be true for precipitation.

- Line 300, incorrect internal reference: should "Chapter 2" be "Figure 1" or "section 2.4.2"?
  The method how the Rossby wave amplitude is computed is described in Chapter 2 (or more precisely in Chapter 2.3). We have changed this and included the reference to Figure 1 (line 313).

- Lines 355-378: the differences between IC-ONLY and the SPP* experiments are small and by eye (Figure 7), do not suggest they are significant. The authors make this point at the end of the paragraph and the section, but only after the reader has read many lines describing minor differences. I propose highlighting the likely lack of significance (and the upcoming section to enhance the ability for statistical testing) earlier in the paragraph and not overstate the differences displayed in Figure 7.
  Thank you for this comment. We followed your suggestion and highlighted the lack of significance and the small magnitude of the signal earlier in the text (lines 371–372). Further, we have included expressions to emphasize that the differences between the experiments are small (e.g. "slightly increased")

- Line 387, typo: should be "3,200' (not '.')
  We have deleted all delimiters for large numbers.

- Figure 8: I wonder if placing all 3 seasons on the same vertical axis would enhance the impression of the larger signal for SON. It looks like the different seasons (in particular, DJF and MAM) would not overlay each other too much; and the relative size of the signals would be much clearer.
  We agree with you that the differences of the magnitude of the signal across seasons would be emphasized if all lines were in one panel, and we had it like this in a first version. However, we decided to split the Figure by seasons, as each panel is easier to read, which becomes difficult when all lines are in one panel (especially with the confidence intervals). We therefore prefer to keep the Figure as it is.

- Lines 443-445: again, this reads as a more general statement on SPP perturbations to individual parametrisations than are shown by the results in this study. Have the authors tested if, for example, the model response is the same from perturbations to the boundary layer scheme (only) and those to the cloud schemes (only)? The study only demonstrates CONV-ONLY and CONV-OFF. More detailed testing can be for a future study, but refrain from making claims that are broader than the scope of this study.
  Unfortunately we have not been able to run further experiments. Especially an experiment with perturbations only to the boundary layer scheme would be very interesting. We have rewritten the sentence such that it is clear that our interpretations are based on the experiments we have conducted (i.e. SPP-CONV-ONLY and SPP-CONV-OFF, see lines 456 ff.). Further, we have added a sentence that additional experiments are required to draw a more general conclusion (lines 460 ff): "To demonstrate this beyond doubt, it would be necessary to conduct additional sensitivity experiments with other configurations of SPP (e.g. with perturbations only to the boundary layer scheme), which is beyond the capabilities of this study."

- Line 465: I couldn't immediately identify where the "10-20%" figure was identified in the earlier results sections. The text references Figures 2 and 9, but it's still not entirely obvious.
  Thank you for this remark. It is true that the exact numbers cannot be easily read from the referenced Figures. However, the point that we are making is somewhat qualitative (i.e. the orders of magnitude are different between the impact on the trajectories and on the Rossby wave amplitude), and we therefore think that a rough indication of the values from different Figures is sufficient.

- Line 478: "likely not of the same order of magnitude as the trajectory count" – I understand that the impact on the two components is likely to be different, but why would the differences likely be different *orders of magnitude*?

  Thank you for this comment, we have rephrased the sentence and hope that the message is clearer now. The sentence reads (lines 493–494): "Therefore, the trajectory count diagnostic might not quantitatively capture the net effect of the diabatically induced divergent outflow on the Rossby wave amplitude."

**Figures**

[Figure]

FIGURE 7 (a) Histogram of global (G) vertical velocities at 500 hPa in bins of width 0.02 Pa·s⁻¹ per forecast, lead time, and ensemble member for the experiments REF, no-SPPT, and no-INI, the unperturbed control member (CF), and the interpolated analysis (ANA). (b) Absolute (solid, left axis) and relative (dashed, right axis) differences between the histograms of REF and no-SPPT in panel (a). Negative (positive) omega values correspond to upward (downward) motion. Note that the left $y$-axis has a linear scale for values between $-1$ and 1, and a log scale for values smaller than $-1$ and larger than 1 [Colour figure can be viewed at wileyonlinelibrary.com]

Figure R1: Figure 7 of Pickl et al. (2022). See the attached caption. Note that experiment REF in Pickl et al. (2022) corresponds to experiment SPPT in this study, and no-SPPT to IC-ONLY.

**References**

Madonna, E., Wernli, H., Joos, H., and Martius, O.: Warm conveyor belts in the ERA-Interim Dataset (1979-2010). Part I: Climatology and potential vorticity evolution, Journal of Climate, 27, 3–26, doi: 10.1175/JCLI-D-12-00720.1, 2014.

Pickl, M., Lang, S. T., Leutbecher, M., and Grams, C. M.: The effect of stochastically perturbed parametrisation tendencies (SPPT) on rapidly ascending air streams, Quarterly Journal of the Royal Meteorological Society, 148, 1242–1261, doi: 10.1002/qj.4257, 2022.

Straus, D. M., Domeisen, D. I. V., Lock, S.-J., Molteni, F., and Yadav, P.: Intrinsic Predictability Limits arising from Indian Ocean MJO Heating: Effects on tropical and extratropical teleconnections, EGUsphere, 2023, 1–29, doi: 10.5194/egusphere-2023-493, 2023.

---

## Author Response (AR2)

**Response to reviewer comments for "Towards a process-oriented understanding of the impact of stochastic perturbations on the model climate"**

**Initial submission on Aug 24, 2023, EGUSPHERE-2023-1938**

Deinhard, M. and Grams, C.M.

Re-submission of revised manuscript on May 12, 2024

Dear Mr Pedram Hassanzadeh, dear reviewers,

thank you for the constructive feedback on our paper. Attached you will find our responses to the reviewer's comments in blue.

**Reviewer 2**

My thanks to the authors for their efforts to address the reviewers' concerns – for both your responses and your edits to the manuscript.

Thank you again for your comments that helped to improve the manuscript!

In particular, thank you for addressing the question of significance of some of the results. I have a few remaining comments on this aspect – one request for the testing CI (Figures 3 & 9) and a query about a potential typo. Other comments are kindly-intended observations:

- Fig 3/9: the thick lines for each experiment line have been used to demonstrate significant difference from 1 (ie different from the IC only experiment). Visually, this is not very easy to distinguish (thick versus thin – are there any thin lines in Fig 9?) and the confidence interval for testing should be stated. As you recognise, this test does not demonstrate how different the MU experiments (Fig 3) are to each other. And I take your argument about interpreting the significance of small differences from 1 being significant as an indication of the likely significance of differences between experiments. Thank you for not over-elaborating on the differences in the text. As a reader, I would remain cautious of some of the smaller differences from what I can see.
  Thank you for this comment. It is true that the thick and the thin lines in Figure 3 and 9 are hard to distinguish. Nevertheless, we kept the Figures as they are, because we think that also other ways of showing the significance (such as marked lines) reduce the readability of the Figure. Regarding the absence of thin lines in Figure 9: Apparently, all values shown in the Figure are significant at the indicated level of confidence. The latter is now specified explicitly in the caption of Figures 3 and 9.

- Figure 9 – you state '1000-sample bootstrapping'. Is this correct? The other tests all claim 10,000-sampling.

Thank you for spotting this, but this is actually not a typo. We chose a lower number of repetitions for the bootstrapping on purpose, as the dataset underlying Figure 9 is by far larger than the other ones. In order to perform the bootstrapping in reasonable amount of time, we therefore decided to go with 1000 iterations only.

- Fig 6/7/8: thank you for adding the confidence intervals. It is striking that here you choose the 10-90% confidence range (as opposed to 95% range for Figure 2). This perhaps reflects that the experiments (or differences between CF and PF in Figure 8) overlap when using 95%. Again, I don't request further changes or response – but this relatively low confidence range (80%, after all) does suggest that one should not read too much into some of the differences: they rather "indicate" something interesting.

  Thank you for this comment, which we agree with. We believe that it would be possible to obtain a more robust signal at higher confidence level with a larger data base, but this was not feasible in this project.

Thanks once again for a very interesting contribution to the literature.